# Effect of the Incorporation of ZIF-8@GO into the Thin-Film Membrane on Salt Rejection and BSA Fouling

**DOI:** 10.3390/membranes12040436

**Published:** 2022-04-17

**Authors:** Elizabeth Gaobodiwe Masibi, Thollwana Andretta Makhetha, Richard Motlhaletsi Moutloali

**Affiliations:** 1Department of Chemical Sciences, Faculty of Science, University of Johannesburg, P.O. Box 17011, Doornfontein, Johannesburg 2028, South Africa; gaobodiwemasibi9@gmail.com (E.G.M.); tamakhetha@uj.ac.za (T.A.M.); 2DSI/Mintek Nanotechnology Innovation Centre—UJ Water Research Node, University of Johannesburg, P.O. Box 17011, Doornfontein, Johannesburg 2028, South Africa; 3Institute for Nanotechnology and Water Sustainability, College of Science, Engineering and Technology, University of South Africa, Florida, Roodepoort 1709, South Africa

**Keywords:** interfacial polymerization, graphene oxide, nanofiltration, thin film composite membranes, zeolitic imidazole framework-8

## Abstract

A series of Zeolitic imidazole framework-8 (ZIF-8) clusters supported on graphene oxide (ZIF-8@GO) nanocomposites were prepared by varying the ratios of ZIF-8 to GO. The resultant nanocomposites were characterized using various techniques, such as Scanning Electron Microscope (SEM), Transmission Electron Microscope (TEM), X-ray diffraction (XRD), Brunauer–Emmett–Teller (BET), thermogravimetric analysis (TGA), Fourier Transform Infrared (FTIR) and Raman spectroscopy. These nanocomposites were incorporated into the thin film layer during interfacial polymerisation process of m-phenylenediamine (aqueous phase which contained the dispersed nanocomposites) and trimesoyl chloride (TMC, organic phase) at room temperature onto polyethersulfone (PES) ultrafiltration (UF) support membrane. The membrane surface morphology, cross section and surface roughness were characterized using SEM and AFM, respectively. Compared to the baseline membranes, the thin film nanofiltration (TFN) membranes exhibited improved pure water flux (from 1.66 up to 7.9 L.m^−2^h^−1^), salt rejection (from 40 to 98%) and fouling resistance (33 to 88%). Optimum ZIF-8 to GO ratio was established as indicated in observed pure water flux, salt rejection and BSA fouling resistance. Therefore, a balance in hydrophilic and porous effect of the filler was observed to lead to this observed membrane behaviour suggesting that careful filler design can result in performance gain for thin film composite (TFC) membranes for water treatment application.

## 1. Introduction

Membrane technology has emerged as a viable, alternate method for wastewater remediation to augment the dwindling fresh water supplies due to its high efficiency and small footprint [1]. The technology has found wide use in desalination, removal of heavy toxic metals and dyes molecules amongst other applications [2]. Reverse Osmosis and Nanofiltration membranes are the most prevalent thin-film composite (TFC) membranes at which the top thin layer may be made of polyamide or polyimide selective layer and porous substrate (polysulfones or polyethersulfone) as the support layer [3]. The polyamide thin film composite (TFC-PA) membranes have been widely used for water treatments due to their wide operating pH range, superior water flux, good stability to biological attack, and good resistance to pressure compaction; however, membrane fouling and chlorine degradation are still considered their main obstacle [4]. Due to these drawbacks, these membranes require extensive pre-treatments to minimize fouling and achieve high permeate quality. In order to reduce the costs and address these issues, scientists have a strong interest in making new membrane materials that will improve fouling resistance, permeability and separation performance [5].

Various methods, such as surface grafting, blending, and coating, have been used extensively. It has been proved that modification of the surface can effectively play a role in reducing chlorination and fouling of PA-TFC membranes since most polymer membranes are sensitive to chlorine. However, the surface modification that is currently used faces great challenges due to decline in permeation of PA-TFC membranes, which is induced by extreme pressure of modification layers. Therefore, the exploration of novel surface modification technologies is of great importance because permeation can be retained while successfully improving the chlorine and fouling resistance of PA-TFC membranes [6]. Current developments in thin-film composite (TFC) membranes for NF and RO applications are aimed at improving the practical performance of the membrane materials over long periods in operations [7]. One of the many ways of improving membrane materials is the incorporation of inorganic fillers which has been demonstrated to address most of these stated challenges [8].

Functional inorganic nanomaterials, such as graphene oxide (GO), zeolites, metal organic frameworks (MOFs), carbon nanotubes (CNTs), etc., are incorporated into the thin polyamine layer to improve its properties [9,10]. Novel inorganic hybrid materials such as zeolitic imidazole framework-8 in combination with graphene oxide have become prominent in the literature in the past few years due to their positive impact on the performance of the resultant membranes [11]. The synergy realised between the hydrophilic GO and porous ZIF-8 are attributed to these gains in observed behaviour [12]. The oxygen-rich graphene oxide not only increases hydrophilicity but also increases the physicochemical properties, such as mechanical strength and thermal stability of the composite membrane [13]. ZIF-8, on the other hand, has high porosity, abundant active surface sites, high surface area and also water stability, making it a favourable candidate for the removal of contaminants in aqueous solutions [14]. The incorporation of MOF and GO can thus greatly improve stability and potential usage of the resultant nanocomposite membranes water quality upgrade [15]. Due to this combination (MOF@GO), advantages such as laminated structure, large pore volumes, alterable pore functionalities, and high surface area can be realised. Moreover, incorporating ZIF-8 particles into GO sheets leads to better control of its properties (structure, morphology, etc.) and eliminates or reduces its aggregation and possible leaching out of the membrane matrix. These are due to the coordination between metal ions and COOH groups of GO which can lead to nucleation of the MOF particles evenly on the surface of GO [16,17].

Herein we report on the incorporation of the ZIF-8@GO nanocomposite in the polyamide layer supported on PES UF support membrane for application in water treatment. It was envisaged that the combination will result in increased water permeation due to the synergistic influence of the hydrophilic GO and porous ZIF-8. The effect of varying ZIF-8 content in the TFN layer on surface fouling, water permeation and solute rejection is reported and interrogated.

## 2. Materials and Methods

### 2.1. Materials

Graphite powder (20 µm, synthetic), orthophosphoric acid (H_3_PO_4_, (85 wt.%), potassium permanganate (KMnO_4_), concentrated sulfuric acid (H_2_SO_4_, 98 wt.%), hydrogen peroxide, ethanol (98 wt.%), diethyl ether, hydrochloric acid (32 wt.%), hexane (95 wt.%), 1,3-phenylenediamine (MPD) and 1,3,5-benzenetricarbonyl trichloride (TMC), zinc nitrate hexahydrate (Zn(NO_3_)_2_.6H_2_O, 98%), methanol (MeOH, 98%) and 2-methylimidazole (2-mIM, 99%) and molecular sieves 4A (purified) used to dry moisture from hexane were all sourced from Sigma Aldrich, Johannesburg (South Africa). Commercial polyethersulfone (PES) (LY PES 100 kDa) sanitary UF membranes used as a support for interfacial polymerization were obtained from Synder filtration, Vacaville (USA). All synthetic reagents were used without any further purification.

### 2.2. Preparation of Graphene Oxide and ZIF-8

GO [18], ZIF-8 [19] and ZIF-8@GO [17] nanocomposites were prepared according to published methods and fully characterised before use. Table 1 indicates the relative amounts of metal, ligand and GO used for the different ZIF-8@GO composite construction.

### 2.3. TFC Membrane Preparation

Polyamide thin composite membranes were synthesized through a modified interfacial polymerization process (Figure 1). Ultrafiltration polyethersulfone (UF-PES) support membranes were pre-treated by soaking in a 0.5% sodium dodecyl sulphate (SDS) solution overnight. The membranes were then washed with distilled water for 1 h, and the pre-treated membranes were immobilized onto glass plates using double-sided tape. Thereafter, aqueous solution of MPD (2% in 100 mL of distilled water at pH of 8 maintained by adding ammonium buffer) was poured onto the top surface of the UF-PES substrate and left to soak for 30 min. Excess amine solution was removed from the membrane surface using a soft rubber roller. Trimesoyl chloride TMC (0.4% in hexane (100 mL)) was poured onto the support UF-PES sheet that was saturated with MPD and left for 60 s. The excess of the organic solution was removed off the surface, and the resulting TFC membrane was cured in the oven at 65 °C for 15 min. The drying step was to provide a further polymerization and to attain the desired stability of the TFC membrane against high pressure. Finally, the resulting membranes were thoroughly washed and kept in DI water until used in carrying out application/performance studies [20].

For the preparation of composite TFC (GO, ZIF-8, ZIF-8@GO) membranes, the same procedure for preparation of TFC was followed, except the addition of ZIF-8@GO (0.1:1, 0.5:1, 0.9:1 and 1.0:1) into the aqueous solution of MPD as illustrated in Table 2. All MPD/additives solutions underwent sonication for 30 min before the interfacial polycondensation reaction was affected [21,22,23].

### 2.4. Characterization of Inorganic Fillers and the PA-TFC Membranes

#### 2.4.1. Fourier Transform Infrared Spectroscopy (FTIR)

GO, ZIF-8, ZIF-8@GO composites (0.1, 0.5, 0.9, 1.0):1, commercial UF-PES membrane and membranes incorporated with different concentrations of ZIF-8@GO were analysed to identify functional groups using a Bruker Vector 22 mid-IR spectroscopy (Bruker, Karlsruhe, Germany) against an air background. Prior analysis, the powder samples of GO, ZIF-8, ZIF-8@GO (0.1, 0.5, 0.9, and 1.0):1 were prepared using 1:9 ratio of a sample and KBr. The membrane samples were placed on the ATR and analysed over the wave number range of 4000–500 cm^−1^.

#### 2.4.2. X-ray Diffraction Spectroscopy (XRD)

X-ray diffraction (XRD) analyses of GO, ZIF-8, ZIF-8@GO composites (0.1, 0.5, 0.9, 1.0):1 were performed at room temperature utilizing a D8 Advance diffractometer (X’Pert, Munich, Germany) with PSD Vantec1 detectors and Cu Kα radiation (λ = 1.5406) source, a tube voltage of 40 kV, a current of 40 mA and an SA 10m slit. The samples were scanned in locked couple mode with 2θ increment in 0.5 s steps. The data obtained were interpreted using high score plus program.’

#### 2.4.3. Raman Spectroscopy (RS)

Raman Micro 200 (Perkin Elmer, Waltham, MA, USA), precisely Spectrometer (Spectrum software), was used to obtain Raman spectra of GO, ZIF-8, ZIF-8@GO composites (0.1, 0.5, 0.9, 1.0) using a laser beam of 5 mW. Prior to analysis, the samples were ground to fine powder, and then placed on a glass plate. The spectra were recorded over a range of 50–3500 cm^−1^ using an operating spectral resolution of 2.0 cm^−1^. The spectra were averaged with 20 scans, at an exposure time of 4 s.

#### 2.4.4. Scanning Electron Microscopy (SEM)

Scanning electron microscopy was used to study the surface morphology of GO, ZIF-8, ZIF-8@GO composites (0.1, 0.5, 0.9 and 1.0) as well as those of membranes incorporated with the composites including the membrane cross-sections. The membranes samples were mounted on a carbon tape and coated with carbon prior to surface morphology analysis. To obtain cross-sectional image analysis, the membrane samples were frozen in liquid nitrogen and fractured whilst hard and finally coated with carbon. SEM micrographs were obtained at an accelerating voltage of 2 kV using a TESCAN Vega TC instrument (VEGA 3 TESCAN software, Brno, Czech Republic), equipped with X-ray detector for energy dispersive X-ray analysis (EDX) operated at 5 kV.

#### 2.4.5. Transmission Electron Microscopy (TEM)

Transmission Electron Microscopy (TEM JEOL, JEM-2010, Akishima, Japan) at an accelerating voltage of 200 kV was used to examine the composite materials (GO, ZIF-8, ZIF-8@GO composites (0.1, 0.5, 0.9, 1.0)). A few milligrams of the samples were sonicated in approximately 5 mL of ethanol using an ultrasonic bath for 10 min. A few drops of the sample specimens were placed on a carbon-coated copper grid and further mounted onto the exchange rod and placed in the TEM chamber for analysis.

#### 2.4.6. Atomic Force Microscopy (AFM)

Atomic force microscopy (AFM, Nanoscale IV, Veeco, Santa Clara, CA, USA) with the spring constant of 0.12 N.m^−1^ through the contact mode in dry air was used to characterize the surface morphology of PES, TFC, and PES/TFC composite membranes. All the membranes were dried for 24 h at room temperature before the AFM analysis was performed. The instrument software was used to obtain roughness factors (R_a_ and R_q_) for the analyses.

#### 2.4.7. Thermogravimetric Analysis (TGA)

The thermal properties and stability of as-prepared samples were determined using a TG-DTA, DT-40 (Shimadzu, Kyoto, Japan) instrument at a heating rate of 10 °C. min^−1^ in the temperature range of 25–800 °C under nitrogen atmosphere.”

#### 2.4.8. Brunauer–Emmett–Teller (BET)

Brunauer–Emmett–Teller (BET) analysis was used to determine the surface area and pore volume of solids. The surface areas and pore volumes of the prepared samples were determined using an automated gas adsorption and surface area analyser, Micrometrics TriStar II Plus Version 3.00 (Micromeritics, Norcross, GA, USA) and Porosity Analyser 3000 (Micromeritics, Norcross, GA, USA). About 0.2 g of the samples were degassed using Micrometrics degassing system at 150 °C in nitrogen at a flow rate of 60 cm^3^ min^−1^ for 4 h.

#### 2.4.9. Contact Angle Measurements

The water contact angle measurements were conducted using the sessile drop method on a contact angle goniometer (G10, KRUSS, Hamburg, Germany). Ten drops of deionized water were deposited on the surface of each membrane, and the contact angle thereof measured to investigate membrane hydrophilicity and hydrophobicity at room temperature. A minimum of ten drop were investigated per sample.

#### 2.4.10. Membrane Performance Measurements Studies

Membrane performance parameters were assessed using pure water flux and solute rejection utilising a dead-end filtration system (Sterlitech Instrument, Kent, WA, USA) under different applied nitrogen gas pressure. The membranes were first compacted with deionized water for 1 h at 1200 Pa prior to flux measurements. Five different pressures (i.e., 700, 800, 900, 1000 and 1100 Pa) were used for the pure water flux Equation (1) studies:(1)$$\left(J_{w}\right)=\frac{\Delta V}{A.\Delta t}$$
where *J_w_* (L.m^−2^.h^−1^) is the pure water flux, *V* is the volume of the permeate (m^3^), *t* is the permeation time (h) and *A* is the effective membrane surface area (0.0013 m^2^).

A conductivity meter was used to measure the salt concentration in the feed and permeate solutions before and after filtration, respectively. The membrane salt rejection was then determined using Equation (2):
(2)$$R(\%)=1-\frac{C_{p}}{C_{f}}\times 100$$
where, *C_p_* is the permeate concentration (ppm) and *C_f_* is the feed concentration (ppm), respectively. After the membranes were subjected to pure water flux measurement (*J_w_*_,1_) for 1 h, a 1000 ppm BSA solution was poured into dead end reservoir and the flux (*J_P_*) of the laden solution was obtained. After 1 h filtration, deionized water was used to backwash membranes for 30 min to remove BSA loose bound on the surface of the membranes thereafter pure water flux (*J_w_*_,2_) was obtained. The flux recovery ratio (*FRR*) was calculated in order to evaluate the fouling-resistant capability of the membrane, using Equation (3). [24]:
(3)$$FRR(\%)=\frac{J_{w2}}{J_{w1}}$$

The total fouling-resistance (*R_t_*) of the membrane was determined using Equation (4) [25]:
(4)$$R_{t}(\%)=\left(1-\frac{J_{p}}{J_{w1}}\right)\times 100$$

Reversible fouling (*R_r_*) and irreversible fouling (*R_ir_*) were obtained using Equations (5) and (6):
(5)$$R_{r}(\%)=(\frac{J_{w2}-J_{p}}{J_{w1}})\times 100$$
(6)$$R_{ir}(\%)=\left(\frac{J_{w1-J_{w2}}}{J_{w1}}\right)\times 100=R_{t}-R_{r}$$

## 3. Results and Discussion

### 3.1. Characterization of GO, ZIF-8 and ZIF-8@GO

#### 3.1.1. Fourier Transform-Infrared (FTIR) Spectroscopy

Figure 1 displays the FTIR spectra of GO, ZIF-8, and the ZIF-8 composites. In contrast to GO alone, after the growth or deposition of ZIF-8 onto GO surface, all composite samples exhibited a characteristic peak at 1727 cm^−1^ assigned to carboxyl C=O stretching band in the ZIF-8 ligand [26]. Moreover, the C-O, O-H, and skeletal C=C vibrations at 1120, 3422, 1648 cm^−1^ are all attributed to GO were observed only in the 0.1:1 composite and were depressed or absent in all the other ZIF-8@GO composite samples. The major absorption bands in the other ZIF-8@GO composites are the vibrational modes emanating from the 2-methylimidazolate ligand at 689, 756, 2928 and 1588 cm^−1^, which are attributed to the aliphatic and aromatic Zn-N, Zn-O, C-H and C=N, respectively [15,27]. The primary absorption bands for ZIF-8 and ZIF-8@GO composites are at 995 cm^−1^, 1145 cm^−1^ and 1309 cm^−1^ corresponding to the C-N bonds in the imidazole group. This imidazolate band is absent in the 0.1:1 composite probably due to the small ZIF-8 content deposited onto the GO surface. Furthermore, the peaks at 757 cm^−1^ corresponds to the Zn-O bonds, and 697 cm^−1^, corresponding to Zn-N bonds, were ascribed to the ZIF-8 structure and these are similar to what Huang et al. observed [26]. The observed FTIR results, therefore, confirm that ZIF-8 nano crystallites were successfully grown or deposited onto GO surface establishing strong interactions between the two components [28].

#### 3.1.2. Scanning Electron Microscope (SEM) and Energy-Dispersive X-ray Spectroscopy (EDS)

The SEM micrographs (Figure 2) show the morphology of the ZIF-8@GO composites. SEM analysis of the 0.1:1 composite (Figure 2a) showed little or no obvious evidence of the presence of ZIF-8 crystallites on the GO surfaces probably because of the small content of ZIF-8 composite present, the only confirmation been the elemental analysis from EDS. In contrast, all composites with higher ZIF-8 content (Figure 2b’–d’) had evenly distributed ZIF-8 crystallites on the GO surfaces as well as higher elemental values in the EDS spectra [29]. Furthermore, (Figure 2a’–d’) showed the relative intensities of the elements (i.e., Zn and N) increased with increasing content of ZIF-8 deposited onto GO support (Table 3). As with the FTIR analyses above, SEM and EDS result confirmed the successful deposition/growth of ZIF-8 on GO support resulting in ZIF-8@GO composites. These observations with respect to relative intensities of the elements is in line with prior reports [30].

#### 3.1.3. Transmission Electron Microscope (TEM)

The exfoliated sheet morphology exhibited by GO in the TEM micrographs (Figure 3a) were in line with expectations [18]. The unsupported ZIF-8 crystallites exhibited hexagonal shape in line with the literature [31]. The ZIF-8 morphology was maintained in the composite materials (Figure 3b–f) demonstrating that its structure was maintained on deposition or growth on GO sheets. The successful growth/deposition of ZIF-8 crystals on the GO surfaces is attributed to hydrogen bonding interaction between the 2-methylimidazole ligand in ZIF-8 and the hydroxyl, carboxyl and epoxy groups present on GO sheets [32]. A secondary growth path is due to the free Zn^2+^ coordinating to the oxygen-containing functional groups of graphene oxide through electrostatic interactions or the metal–oxygen covalent prior to coordinating with the ligands leading to fast self-assembly onto GO sheets leading to ZIF-8 crystallite growth [15,30].

#### 3.1.4. X-ray Diffraction (XRD) Analysis

Powder XRD analyses were done to confirm the crystal structure of GO and its hybrid materials containing ZIF-8. The diffraction patterns (Figure 4) for the fabricated materials indicate that they were all crystalline in nature as evidence through the sharp diffraction bands. As expected, the XRD pattern of GO exhibited an intense peak around 8.7° with no evidence or hint of the starting graphite material. On the other hand, intense peaks were found at 2θ of 7.6, 10.8, 12.9, 14.8, 16.9, 18.1, 24.5, 26.6 for ZIF-8 corresponding to (110), (200), (211), (220), (310), (222), (233) and (134) planes in line with the prior literature reports [26,33]. These diffractions were maintained in the composites, ZIF-8@GO, composites, albeit with a slight shift to higher values (see the 200 band, Figure 4), indicating that the ZIF-8 crystalline structure was maintained with growth on the GO surface [31]. The presence of GO diffraction gradually decreased/disappeared with increasing ZIF-8 content demonstrating total delamination of the sheets with increasing ZIF-8 content. At 0.1 wt.% ZIF-8, no characteristic peaks of ZIF-8 was observed presumably due to the small content similar to other prior studies [28]. Therefore, XRD also confirms that the nanocomposites were successfully grown on the GO surface, in agreement with the other complementary techniques discussed earlier.

#### 3.1.5. Raman Analysis

Figure 5 shows Raman spectra of GO, ZIF-8 and the four ZIF-8@GO composites. The G- and D-band of GO appeared at 1584 cm^−1^ and 1339 cm^−1^, respectively, in line with expectations [32,33]. The Raman spectrum of ZIF-8 had bands at 647, 692, 843, 957, 1031, 1155, 1193, 1393, 1467 and 1516 cm^−1^ as expected, which were assigned to the methyl group and vibrational modes of imidazole ring [27,34] and those at 1467, 1155 and 692 cm^−^^1^ assigned to methyl bending, C−N stretching and imidazolium ring puckering, respectively [28]. In contrast to baseline GO, the Raman spectra of the four composites show that as the content of ZIF-8 increased, there was a concomitant shift in D and G bands as well as a drastic intensity increase for the D band. For instance, the D band relative intensity increased from 728 a.u. (0.1:1), 782 a.u. (0.5:1), 975 a.u. (0.9:1) to 4718 a.u. (1.0:1) with increasing ZIF-8 content. This observation is ascribed to the decrease in the mean size of the sp^2^ domains upon the increase in ZIF-8 content. The ratio of the intensity of the D and G band (I_D_/I_G_) showed that as the content of ZIF-8 was increased in the composite, the value of I_D_/I_G_ for ZIF-8@GO (0.1, 0.5, 0.9, 1):1 also increased (Table 4) thereby confirming that ZIF-8@GO composites were successfully synthesized [35]. The growth of this ratio suggests that the amount of defects consequently increased.

#### 3.1.6. Brunauer–Emmett–Teller (BET)

The nitrogen adsorption-desorption isotherms for GO, ZIF-8 and ZIF-8@GO composites from which the BET surface area, pore volume, and pore size were obtained are shown in Figure 6. ZIF-8@GO composites showed a type IV isotherm with type H3 hysteresis loop [36]. It was observed that as the content of ZIF-8 on GO surface increased, the BET surface area increased progressively whilst the pore size and pore volume decreased (Table 4) in agreement with previous reports [14]. GO exhibited a lower surface area of 21.93 m^2^.g^−1^ compared to ZIF-8 which had a much highest surface area of 985.37 m^2^.g^−1^. The surface area of the composites was found to be slightly lower than the surface area of self-standing ZIF-8 because of the presence of the lower surface area GO leading to an overall decrease in the surface area of the composites. The mesoporous nature together with the improved specific surface area makes ZIF-8@GO composites more ideal materials for water purification application [37]. The variations in the physicochemical characterization observed here are an indirect, positive indication that the nanocomposites agree with other techniques above.

#### 3.1.7. Thermogravimetric Analysis (TGA)

Thermograms for GO, ZIF-8 and ZIF-8@GO composites obtained under a N_2_ atmosphere all exhibited three thermal events (Figure 7). For GO, the initial weight loss gradually started from 25 °C to beyond 180 °C hence its accelerated for the second phase, from 180 °C to 300 °C attributed to decomposition of the functional groups, i.e., carboxyl, epoxy, and hydroxyl groups, on the GO sheets. The third phase between 300 °C and 400 °C was due to the decomposition of GO hexagonal carbon skeleton [14]. For ZIF-8 was observed to have small mass loss, less than 10% overall, up below 300 °C attributed to its structural collapse and decomposition of 2-mIM [38,39]. This quantum and sequence of the weight decrease was related to the content of ZIF-8@GO composites present in line with prior reports. It was not obvious why the decomposition of ZIF-8@GO 0.9:1 differs with those of the other composites, previously Chu attributed such deviations to the loss of guest molecules such as moisture not adequately release during drying [28]. The increased thermal stability observed for the composites is reflective of the strong interaction between the ZIF-8 and GO units [40,41].

### 3.2. Characterization and TFC Membrane Performance

#### 3.2.1. FTIR Ccharacterizations of the PES (M0), PA-TFC (M1), GO (M2), ZIF-8 (M3), ZIF-8@GO (0.1:1) (M4), ZIF-8@GO (0.5:1) (M5), ZIF-8@GO (0.9:1) (M6), ZIF-8@GO (1:1) (M7) Membranes

Figure 8 showed the FTIR spectra of all the membranes (M0-M7) while Figure 8b is the expansion of specific regions for selected membranes for enhanced clarity. The M0 characteristic bands were found at 1585, 1493 and 1240 cm^−1^ attributed to the aromatic band, C-C stretching, a benzene ring, and aromatic ether band functional groups [42,43]. Similar bands were also observed for all other TFC membranes, which were coupled with the emergence of new bands at ca. 620 and 1320 cm^−1^. These new bands were assigned to vibrations of the phenyl ring and C-N stretching vibrations [44]. The C-N stretching vibrations correspond to the amide structure of the PA layer during IP and hence confirmation of the formation of the thin film [45]. Moreover, the peak intensity with the slight enhancement found at 1662 cm^−1^ may be allocated to the new amide linkages formed by the reaction of the –NH_2_ groups in MPD with –COOH groups in GO [46]. In addition, with the incorporation of GO, the peak intensity increment observed at 3061 and 3092 cm^−1^ was due to symmetric and asymmetric stretching vibrations of additional C-H bonds. The FTIR spectra of ZIF-8 showed the most Zn–N stretch mode was detected at 420 cm^−1^ [47]. Upon the incorporation of ZIF-8@GO, additional bands that were observed were attributed to the imidazole groups indicative of the presence of ZIF-8@GO composite in the thin film layer. However, the majority of the peaks are masked by PES peaks due to their minute amount in the membranes [44].

#### 3.2.2. SEM Surface Morphology of Membranes

The smooth surface morphology of the pristine PES membranes was dominated by large micropores as expected (Figure 9) [48]. The smooth membrane surfaces became rougher as the thin film layer formed due to the reaction of the monomers, MPD (2 wt.%) and TMC (0.4 wt.%), through interfacial polymerization. The surface roughness progressively increased further with the addition of nanofillers into the thin film (Figure 9c–h), accordingly, i.e., for GO, ZIF-8, ZIF-8@GO (0.1:1, 0.5:1, 0.9:1, 1.0:1) [49]. Importantly, these TFC membranes did not exhibit any noticeable pores compared to the base membrane even as the nanofillers were increased (Figure 9a) further confirming presence and integrity of the thin film polyamide layer on the support membrane [50].

#### 3.2.3. Cross-Section Analysis of Membranes

The cross-section of the membranes presented in Figure 10 showed a dense skin layer which is connected by a porous sub-layer having finger like pore structures. The formation of polyamide layer did not temper with the finger-like pore structures hence the structures can be observed in all membranes [51,52,53].

#### 3.2.4. AFM Surface Analysis of the Membranes

The roughness of the membrane surface was obtained using AFM (Figure 11) and related parameters calculated and presented in terms of R_a_ and R_q_ (where R_a_ is mean roughness and R_q_ is the root mean square) in Table 5. PES base UF membrane (M0) exhibited the least number of nodules like surface and had the highest surface roughness. Upon the formation of the polyamine layer on top of the support layer, the surface roughness seemed to have decreased as the nodule-like structures start to shrink in line with expectations [54]. However, with increasing nanomaterials embedded, the nodule-like structures disappear, and the formation of sharp peaks started to appear which is reflected in the slight increase in the roughness parameters (Table 5) as the content of the composites increased, ZIF-8@GO (0.1:1, 0.5:1, 0.9:1 and 1.0:1). The significant decrease in R_a_/R_q_ was observed for M0 from 215.40/278.35 to M4 with 46.91/64.48 since it contained more of GO [55].

#### 3.2.5. Water Contact Angle (WCA)

The relative hydrophilicity, as represented by the water contact angle, of all the membranes (M0–M7) is presented in Figure 12, with higher values indicative of hydrophobic character. All the thin-film membranes had a lower contact angle compared to the baseline UF PES support reflecting their higher relative hydrophilicity (82°) [56]. Three reference TF membranes are included for the purpose of elucidating the effects of ZIF-8 and GO. TFC membranes containing composites with more GO character, e.g., with the lowest content of ZIF-8 (namely M2 and M4 membranes), exhibited the lowest water contact angle indicating that the oxygen functional groups in GO significantly enhanced the hydrophilicity of TFC membranes [33,57]. In general, however, the membranes with GO or ZIF-8@GO composites showed a decreasing WCA with increasing content of the nanofiller. This is attributed to the increased density of carboxyl, epoxy and hydroxyl groups being exposed on the surface of the membrane as filler content increased. As the character of the nanofiller became more ZIF-8 like, for instance, when ZIF-8 content is increased at constant GO (M4, M6 and M7), the WCA was negatively affected (Figure 12). This is probably due to decrease in overall hydrophilic character (Figure 12) as GO is fully covered with growing ZIF-8 content (Table 4). However, the CA for M5 was higher than all ZIF-8@GO composite membranes even though the content of ZIF-8 was less of M6 and M7. This could be due to the defects formed on the thin layer during IP. Since ZIF-8 is known to be a hydrophobic type metal organic framework, this observation is thus justified [25].

#### 3.2.6. Water Flux

The pure water permeation at different applied pressures is presented in Figure 13. The flux seems to track the WCA measurements, i.e., the membranes with lower WCA presented higher pure water flux in line with prevailing hypotheses and knowledge. Furthermore, as the applied pressure was increased, pure water flux for all membranes (PES, GO, ZIF-8 and different ZIF-8@GO loadings) increased as expected. The permeability (Table 5), which is calculated from the slope of pressure v/s flux, was dramatic for the TFC (highest of 0.27 v/s 7.9 L.m^−2^h^−1^) as compared to the base membrane (lowest of 0.036 v/s 0.1 L.m^−2^h^−1^) indicating that base UF membranes have been transformed into NF type membrane on deposition/growth of thin film composite layer. This was in line with the observations of Shen et al. [57], whereby it was discovered that the content of GO had an effect on high fluxes of TFC membranes. The differences in the flux performance of the membranes containing different types of fillers, that is, GO, ZIF-8 and ZIF-8@GO composites, is attributed to the overall TFC membrane hydrophilicity and additional water flow pathways afforded through the porous nanofillers [53]. The observed flux behaviour is in line with previous reports when fillers were introduced through the MPD component in the support UF membrane [58]. Thin film membrane without fillers gave the least water permeability of 2.8 L.m^−2^h^−1^ (at the highest applied pressure) compared to modified TFC membranes which reached up to 7.9 L.m^−2^h^−1^ at highest pressure (1100 KPa) used in line with reported findings [50].

Membranes which were dominated by GO character (with less amount of ZIF-8), e.g., M2 (GO) and M4 (ZIF-8@GO 0.1:1), had the highest flux due to the following; (I) the addition of GO into thin film layer increased the hydrophilicity of the membrane surface, which can draw water molecules into the matrix through to the pores of the support UF membrane and thus aiding the water transportation through the membrane [5], (II) the addition of GO (hydrophilic properties which attract water molecules) increase mass transport resistance between the active and supporting layers contributing to an enhanced water flux [59], (III) the presence of inorganic filler-organic polymer matrix (GO/PA) discontinuity interphase defects [57]. The addition of porous ZIF-8 leads to enhanced water transport or passage as it lowers the tortuosity of the matrix.

#### 3.2.7. Rejection of Salts

Solute rejection studies were assessed using three salts, viz. NaCl, MgCl_2_ and Na_2_SO_4_ at 1000 ppm concentration. Figure 14 presents the membrane salt rejection performance of all the membranes (M0–M7) obtained at applied pressure of 900 KPa. The results indicate that the modified membranes (PA-TFC) gave better rejection (ranging from 45 (M1 for NaCl) to 98% (M5 for Na_2_SO_4_)) for the salts compared to unmodified PES membrane (ranging from 40–50%) in line with expectations [60]. The divalent Mg ion with relatively high charge means it is strongly attracted to the negatively charged membrane surface than the monovalent Na ion, resulting in the different observed rejection profiles [61]. Jamil et al. also reported results confirming the low rejection for monovalent ions and high rejection of divalent ions while using NF membranes [51]. The rejection performance of PA-TFC membranes to salt is controlled by both the Donnan and size exclusion effect where the surface of the membrane with negative charge has better salt separation for monovalent cations and divalent anions than divalent cations and monovalent anions [51,62]. The fabricated TFC membranes exhibited the highest rejection for salts containing the polyatomic divalent anion SO_4_^2−^ (98% for M5 with Na_2_SO_4_) compared to monoatomic anions Cl^−^ ions (from both NaCl and MgCl_2_). This was ascribed to the combination of physicochemical properties of dense PA composite layer, i.e., size exclusion [5] as well as the effect of the strongly negative charge of membrane surface towards divalent charge. In line with this, the rejection decreased when the monoatomic monovalent anion Cl^−^ was assessed, i.e., smaller, and less charged Cl^−^ experienced relatively less resistance to pass than SO_4_^2−^. It was also observed that as the concentration of ZIF-8 was increased in the ZIF-8@GO composites (M4, M5, and M6) the salt rejection increased for all the salt used (both mono and divalent ions). This might be due to addition interaction forces introduced through the ZIF-8 component, tortuosity and interactions as solutes pass within the porous structures as well as adsorptive interactions of nanocomposite fillers. However, the decline was observed for M3 as compared to other membranes whereby only ZIF-8 was used as a filler. This might be due to the absence of GO which provide the membrane with more negatively charged functional groups that provides the membrane with separation properties (Donnan exclusion model) [63]. The trend in salt rejection was, in the diminishing order: Na_2_SO_4_ > MgCl_2_ > NaCl. Therefore, the Na_2_SO_4_ solution containing ions with relatively higher valence and size (Mg^2+^ or SO4^2−^) ions were rejected more efficiently (>90% for NaSO_4_ and >80 for MgCl_2_) by the TFN membranes compared to solutions with smaller, monovalent chloride anions (NaCl at ca. 50%) [48]. This observed salt rejection behaviour is typical of TFC membranes and therefore clearly indicate that the fabrication of the targeted membrane type as well as behaviour was realised.

#### 3.2.8. Membrane Fouling Assessment

Membrane fouling remains one of the biggest obstacles for efficient operation of membranes. Fouling propensity or antifouling behaviour of PES support membrane, PA-TF membrane and PA-TFC membranes containing GO, ZIF-8 or ZIF-8@GO were evaluated by measuring the recovery of pure water flux before and after the fouling of the membrane with 1000 ppm of Bovine Serum Albumin (BSA) solution. In the first instance, flux of BSA containing solution was measured over time, 175 min (Figure 15). Here it was observed that the support PES membrane had a relatively faster permeate flux decline (~50%) when compared with the PA-TF membranes which all had a slower flux decline (<10%) in the same period. This indicated that the combination of a more porous and relatively hydrophobic surface resulted in higher affinity and hence clogging of pores by BSA molecules in line with prior reports [33]. On the other hand, the PA-TFC membranes containing GO alone showed a higher flux decline (~12%) than that containing ZIF-8 alone (M3, <5%) or the reference PA membrane (M1, <5%). The variation of the different composites tracked those which they resemble closely, i.e., those with low ZIF-8 behaved closer to the GO filler alone (M2) and those with higher ZIF-8, closer to the ZIF-8 filler alone (M3), albeit the differences were small.

Secondly, membrane fouling behaviour was also assessed using flux recovery ratio (FRR) (Figure 16). The pristine membrane, M0, showed the lowest FRR of all the membranes indicative of relatively higher irreversible BSA fouling. The reference PA-TF (M1) membrane also exhibited higher irreversibility at 50%. PA-TFC containing only GO showed the highest FRR (85%) together with the composite with the least amount of ZIF-8 (M4) at 88%. Nonetheless, all the composite membranes containing ZIF-8@GO had FRR above 70% confirming their high fouling resistance as seen earlier.

In addition to FRR, membrane fouling on the surface or inside its pores is also assessed using reversible, irreversible and total fouling calculations [64]. These parameters are indicative of how foulants interact with or attached to the membrane. Reversible fouling occurs when foulants are loosely bound to the membrane and therefore can be easily removed through a sufficient shear force or backwashing. However, it is a different case with irreversible fouling since the fouling agents are tightly attached to the membrane surface and can only be removed by chemical cleaning [65,66,67]. Figure 17 represents reversible fouling ratio (*R_r_*), irreversible fouling ratio (*R_ir_*) and total fouling ratio (*R_t_*) for all the fabricated membranes. These results revealed that R_t_ of the support PES membrane is higher than those of the PA-TFC membranes. The *R_ir_* of PES is much higher (68%) whereas those of PA-TFC membranes are all below 45% with the lowest at 12% (M4).

These results demonstrated that the nanocomposites (ZIF8@GO) imparted a positive effect on BSA fouling response. The surface hydrophilicity (Figure 12) played an important role in the lowering the BSA adsorption affinity of the membranes. Breite attributed this effect to the presence of multitudes of functional groups present on both GO and ZIF-8 fillers in the selective layer of the PA-TFC membranes [68].

## 4. Conclusions

The thin film polyamide layer was successfully grown on the surface of PES support substrate via interfacial polymerization method. Characterization’s techniques such as (SEM-EDX, AFM, ATR-FTIR) confirmed the formation of the polyamide thin layer upon MPD and TMC loading. The techniques also revealed that the modified membrane composites possess ridge-valley and noodle morphology with better surface roughness than pristine PES. Contact angle, water intake capacity and flux analysis revealed that varying concentrations of composite nanomaterials have a positive impact on the membrane hydrophilicity of the membrane. The TFC membranes incorporated with GO, ZIF-8, ZIF-8@GO composites displayed a better flux and rejection performance because of their exceptional properties. The negatively charged PA-TFC/GO and ZIF-8@GO membranes experienced the highest rejection of more than 90% for multivalent SO_4_^2−^ as compared to monovalent NaCl and divalent MgCl_2_ which is ascribed to the combination of physicochemical properties of dense PA layer, i.e., size exclusion, and negative charge on the membrane surface. The decreased rejection observed for NaCl and MgCl_2_ solutions might be due to their relative smaller sized ions as well as lower change leading to a higher permeation of the solute. Membranes with lower surface roughness displayed a better fouling propensity than pristine PES and TFC membranes.

## Data Availability

Not applicable.

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
