# Peer review of "Effect of the Incorporation of ZIF-8@GO into the Thin-Film Membrane on Salt Rejection and BSA Fouling"

_membranes, 2022, doi:10.3390/membranes12040436_

Round 1

Reviewer 1 Report

Please see attached word document, thanks

Author Response

Responses attached

Reviewer 2 Report

The purpose of the manuscript is to present the methods for modifying TFC RO membranes with ZIF8-GO particles at different compositions and evaluate the surface properties as well as flux, salt rejection and protein fouling studies. Overall, the authors had done extensive characterization and optimization of their membranes. The paper is generally well-written with the conclusions supported by adequate data. There are some minor clarifications and corrections that need to be addressed. i recommend that the manuscript be published once the issues below are addressed. 

  1. page 2, authors have stated in the 3rd paragraph that "ZIF-8 on the other hand with its high porosity, abundant active surface sites and high surface area, make it a favourable candidate to get rid of pollutants in wastewater." This statement is not accurate. Porosity. Can authors elaborate on how high porosity, active surface sites and high surface area  result in pollutant removal?
  2. Table 1, column 2, Please specify what this ratio 
  3. Figure 2 captions, Please specify the ZIF-8/GO ratios associated with each of the images,for e.g. (a) and (a;) correspond to 0.1:1, etc. 
  4. Figure 3, Please be consistent with the scale label. Either use 0.5 um or 500 nm
  5. Figure 9, page 17. The authors may want to standardize the magnifications of all the membranes surface for easier comparison. The last image (h) is of different magnifications compared to the others.
  6. Section 3.2.3, page 17. The authors may wish to explain in greater details how they are able to determine the decrease of microvoids from the images. Microvoids are generally not observable from the SEM images. Are the authors referring to macrovoids? However, the formation of polyamide layer will not affect the amount of macrovoids.
  7. Table 5, page 19. Could the authors kindly explain how the permeability in table 5 is obtained and the authors may wish to explain how the permeability is related to the surface roughness. There may be a typo in the title.
  8. Figure 11, Please indicate the height scale for all AFM images
  9. Figure 14, page 23. The authors will want to verify if their thin film composite membranes M1 is defect-free as it is unlikely that based on the formulation, the TFC membrane only haS a slightly higher rejection (about 50%) thaN the PES substrate which itself has a NACl rejection of about 40%.
  10. Please review the manuscript for grammatical and typographical errors. 

Author Response

Responses attached
